



# iFit: An intensity based retrieval for volcanic $SO_2$ from scattered sunlight UV spectra

Ben Esse[1], Mike Burton[1], Matthew Varnam[1], Ryunosuke Kazahaya[2], and Giuseppe Salerno[3]

[1]School of Earth and Environmental Sciences, The University of Manchester, Manchester, M13 9PL, UK
[2]Japan Meteorological Agency, Tokyo, Japan
[3]Istituto Nazionale di Geofisica e Vulcanologia, Osservatorio Etneo, sezione di Catania, Piazza Roma 2, 95123, Catania, Italy

**Correspondence:** Ben Esse (benjamin.esse@manchester.ac.uk)

**Abstract.** Accurate quantification of the sulphur dioxide ($SO_2$) flux from volcanoes provides both an insight into magmatic processes and a powerful monitoring tool for hazard mitigation, with miniature ultraviolet spectrometers becoming the go-to method for $SO_2$ flux measurements globally. The most common analysis method for these spectrometers is Differential Optical Absorption Spectroscopy (DOAS), in which a reference spectrum taken outside the plume is used to quantify the $SO_2$ column density inside the plume. This can lead to problems if the reference spectrum is contaminated with $SO_2$ as this leads to systematic underestimates in the retrieved $SO_2$ column density. We present a novel method, named "iFit", which retrieves the $SO_2$ column density from UV spectra by directly fitting the measured intensity spectrum using a high resolution solar reference spectrum. This has a number of advantages over the traditional DOAS method, primarily by eliminating the requirement for a measured reference spectrum. We show that iFit can accurately retrieve $SO_2$ column densities in a series of test cases, finding excellent agreement with existing methods without the use of a reference spectrum. We propose that iFit is well suited to application to both traverse measurements and permanent scanning stations, and shows strong potential for integration into volcano monitoring networks at observatories.

## 1 Introduction

Measuring the flux of sulphur dioxide ($SO_2$) from volcanoes is an extremely valuable tool in volcanology for investigating magmatic processes (Delgado-Granados et al., 2001; Fischer et al., 1994; Oppenheimer et al., 2011; Salerno et al., 2018), volcano monitoring (Sparks, 2003) and quantifying global fluxes of volcanic gases (Burton et al., 2013). $SO_2$ is often easier to measure than other more abundant gases in volcanic plumes (such as water or carbon dioxide) for two reasons - it has a strong absorbance signature in both the ultraviolet (UV) and infrared (IR) regions of the electromagnetic spectrum and it is not usually present in the atmosphere. These features lead to the widespread use of correlation spectroscopy (COSPEC) to determine $SO_2$ fluxes at various volcanoes (Caltabiano et al., 1994; Moffat and Millan, 1971). The development of miniature UV spectrometers further revolutionised $SO_2$ flux quantification (Galle et al., 2002; Kantzas and McGonigle, 2008; Kantzas et al., 2009), and these have now become the main tool worldwide for monitoring volcanic gas fluxes (Burton et al., 2009; Edmonds et al., 2003; Galle et al., 2010; Martin et al., 2010; Salerno et al., 2009a, b). These spectrometers typically measure the spectrum of scattered UV sunlight that passes through the volcanic plume, from which the $SO_2$ column density is retrieved





using Differential Optical Absorption Spectroscopy (DOAS) (Platt and Stutz, 2008). The widespread use of DOAS has been further cemented by its ability to detect and quantify other volcanic gases, such as BrO (Bobrowski and Platt, 2007; Bobrowski et al., 2003), $H_2S$ (O'Dwyer et al., 2003), OClO (General et al., 2015) and recently $H_2O$ (Kern et al., 2017). There are various software packages that allow DOAS analysis to be performed, such as DOASIS (Kraus, 2006), QDOAS (Danckaert et al.,

2017) and UVolc (Kantzas et al., 2012).The introduction of $SO_2$ cameras, which allow $SO_2$ column images to be collected, has further diversified the use of UV spectrometers as they are often used to provide the necessary $SO_2$ calibration (Burton et al., 2015). Satellites may also be used to measure $SO_2$ flux time series when combined with weather data (Carn et al., 2017; Pardini et al., 2017, 2018), and the validation of such measurements is only possible when accurate ground-based flux data are available.

Though DOAS is a powerful technique, it requires a measured reference spectrum outside the volcanic plume (with no $SO_2$ present). This is not always easy to achieve, as the plume can be very wide or can move due to changing wind direction during measurements. If the reference spectrum is contaminated with $SO_2$ this will lead to a systematic offset in the $SO_2$ retrieved in the plume. The reference spectrum also requires updating periodically throughout measurements as the background lighting conditions change over time. To reduce these errors a number of methods have been developed that use a synthetic reference

spectrum, especially for permanent scanning stations (Salerno et al., 2009b; Lübcke et al., 2016). Although this eliminates the problem of $SO_2$ contamination, the reference spectrum used in the retrieval is still of a low resolution (that of the spectrometer) and so does not properly represent the natural sunlight spectrum. This is known as the $I_0$ effect, and corrections must be performed to account for this (Platt and Stutz, 2008, Section 6.7).

Here we present a new method, named "iFit", for retrieval of volcanic $SO_2$ through a direct fit of the measured intensity

spectrum using a forward model built on a high resolution Fraunhofer spectrum. This allows for determination of the $SO_2$ column density without the need for a plume-free reference spectrum. As with other methods using a synthetic reference iFit will never suffer from the issue of $SO_2$ contamination, but as the forward model calculations all take place on the high resolution model grid there is no need to correct for the $I_0$ effect.

This paper is structured as follows. We begin by explaining the important instrumental effects that must be considered in a

UV spectral fitting procedure (section 2.1), followed by an overview of the DOAS methodology (section 2.2) and a detailed discussion of the iFit procedure (section 2.3). We then present several test cases, including measurements of clear sky (section 3.1), $SO_2$ calibration cells (section 3.2) and the plume of Masaya volcano in Nicaragua (section 3.3). Finally we investigate the various benefits of using a fully synthetic forward model and discuss the future development and application of this novel technique. Due to its great flexibility and accuracy iFit is a powerful tool for volcano monitoring at observatories and for field

volcanologists. We therefore attempt to present the technique in a manner that allows the widest possible audience to use it.





## 2 Retrieval of volcanic SO$_2$ from UV spectra

### 2.1 Instrumental Effects

Miniature UV spectrometers have become the go-to method for measuring volcanic SO$_2$ fluxes. They are small, lightweight and relatively cheap, making them ideal for monitoring volcanoes. iFit was developed using Ocean Optics (http://www.oceanoptics. com) USB2000, USB2000+ and Flame spectrometers, but the principles will apply to other UV spectrometers.

Most instrumental effects that impact DOAS retrievals also affect iFit. These include the wavelength calibration and resolution of the spectrometer and the presence of the dark current, bias signal and stray light in the measured spectrum. One effect that does not require correcting in DOAS is the pixel-to-pixel variation in sensitivity to light, which we call the "flat spectrum". Due to slight differences in the manufacturing process each individual pixel in the CCD array has a slightly different quantum efficiency, and so will record a different intensity when illuminated by a spectrally flat light source. This effect is a property of individual pixels and has no dependence on the wavelength of light illuminating the pixel, meaning it is not affected by any changes to the wavelength calibration of the spectrometer.

The flat spectrum can be measured using a smoothly varying light source. The measured signal is averaged across many spectra to minimise noise and normalised with either a fitted polynomial or a boxcar smoothed spectrum to remove the broad features of the lamp, leaving the flat spectrum. This spectrum is different for each spectrometer, but once measured it should remain usable for extended periods of time (Fig. 1).

### 2.2 DOAS

Currently, the most widely used technique for volcanic SO$_2$ retrievals is DOAS, which utilises the Beer-Lambert Law to retrieve gas column amounts. A full description is given by Platt and Stutz (2008), but a brief overview of the key points will be provided here in order to highlight the differences between DOAS and iFit. In DOAS the measured plume spectrum, $I(\lambda)$, is divided by a reference spectrum taken outside the plume, $I_0(\lambda)$, to produce a transmittance spectrum. The optical depth, $\tau(\lambda)$, is then found by taking the negative natural logarithm of the transmittance spectrum. The gas column density is related to the optical depth:

$$\tau(\lambda) = -\ln\left[I(\lambda)/I_0(\lambda)\right] = \sum_i [\sigma_i(\lambda) \cdot a_i] \tag{1}$$

where $\sigma(\lambda)$ is the wavelength dependant absorption cross section, $\alpha$ is the total column gas amount, and $i$ represents the individual absorbing species. The individual gas column densities can then be retrieved by varying $\alpha$ until the optical depth spectrum matches that measured by the spectrometer. One of the benefits of DOAS is the ability to measure multiple gas species simultaneously. Note that both $I(\lambda)$ and $I_0(\lambda)$ are measured by the same spectrometer, so the flat spectrum is automatically corrected for in DOAS retrievals.





Equation 1 describes the case of an absorbing gas – however in reality there are a number of processes contributing to the actual optical depth, including Rayleigh and Mie scattering. These are added to equation 1 to give:

$$\tau(\lambda) = \sum_i [\sigma_i(\lambda) \cdot a_i] + \epsilon_R(\lambda) + \epsilon_M(\lambda) \tag{2}$$

where $\epsilon_R$ and $\epsilon_M$ are the Rayleigh and Mie scattering optical depths respectively. In DOAS retrievals these are described by a single polynomial term, $P(\lambda)$:

$$\tau(\lambda) = \sum_i [\sigma_i(\lambda) \cdot a_i] + P(\lambda) \tag{3}$$

These broadband features can either be included in the spectral fitting process alongside the gas amounts or removed from the measured optical depth spectrum prior to the fit, thus allowing the broadband features to be ignored. This is achieved by high pass filtering the optical depth spectrum, usually by either fitting a polynomial of a suitably high order or by removing a smoothed spectrum. The same process is applied to the absorption cross-section, giving the following equation:

$$\tau'(\lambda) = \sum_i [\sigma_i'(\lambda) \cdot a_i] \tag{4}$$

where the dash denotes a differential spectrum. This allows the gas column amount in a measured spectrum to be retrieved. For the remainder of this section we will assume the broadband features have been removed.

20 **2.2.1 Spectral Resolution**

In most cases the spectrometers cannot fully resolve the absorption lines of the gas, so this must be taken into account before analysing spectra. This is achieved by convolving the gas absorbance spectrum with the spectrometer Instrument Line Shape (ILS), which modifies equation 4 to give:

$$\tau'(\lambda) = \sum_i [ILS \otimes (\sigma_i'(\lambda) \cdot a_i)] \tag{5}$$

Where $\otimes$ denotes a convolution. The ILS used can either be a mathematical function (such as a Gaussian) or be measured directly from the spectrometer using a spectral line source (e.g. a mercury lamp). It is important to note that convolutions and multiplications are non-commutative, so that:

30 
$$ILS \otimes (\sigma_i'(\lambda) \cdot a_i) \neq (ILS \otimes \sigma_i'(\lambda)) \cdot a_i \tag{6}$$





This means that the convolution should be applied to each gas spectrum included in the retrieval for every iteration in the fitting procedure. The shape of the ILS can change due to ambient temperature fluctuations, so it is best to stabilise the temperature of the spectrometer (Pinardi et al., 2007). This is not always practical, in which case the effect of the ILS should be considered
in the final retrieved $SO_2$ uncertainty budget.

### 2.2.2   The $I_0$ effect

Related to the spectrometer resolution is the $I_0$ effect, which affects DOAS retrievals but not iFit (see section 2.3). Due to the finite resolution of the spectrometer some of the information in the natural spectrum is lost. This means that the measured optical depth spectrum does not truly represent the actual optical depth:

$$\tau^*(\lambda) = -\ln\left[I^*(\lambda)/I^*_0(\lambda)\right] \tag{7a}$$

$$\tau(\lambda) = -\ln\left[I(\lambda)/I_0(\lambda)\right] = -\ln\left[(ILS \otimes I^*(\lambda))/(ILS \otimes I^*_0(\lambda))\right] \tag{7b}$$

$$\tau(\lambda) \neq \tau^*(\lambda) \tag{7c}$$

where the * represents a natural spectrum. To correct for this effect a modified absorption cross-section is used (Aliwell et al., 2002, Appendix 2). This corrected cross-section is calculated as:

$$\sigma_{corr}(\lambda) = -\ln\left[(ILS \otimes (I^*_0(\lambda) \cdot \exp[-\sigma(\lambda) \cdot \alpha_c]))/(ILS \otimes (I^*_0(\lambda)))\right]/\alpha_c \tag{8}$$

In this equation, a high resolution Fraunhofer spectrum, $I_0{}^*$, is multiplied by a transmittance of a gas with column density $\alpha_c$ and convolved with the spectrometer ILS. This is then normalised by an ILS convolved Fraunhofer spectrum with no gas absorption to produce a synthetic plume transmittance spectrum, which is then converted to absorbance by taking the negative natural logarithm. The corrected cross-section is found by rearranging equation 1.

The choice of the column density used here is important. This corrected cross-section is only true for the exact value of
$\alpha$ chosen, and becomes progressively worse the further away the actual column density is from the chosen value (common practice is to set $\alpha_c$ to be the highest expected $SO_2$ amount). This allows for another potential source of user-induced error in retrievals.

### 2.2.3   The Ring effect

Inelastic scattering in the atmosphere leads to an observed infilling of the Fraunhofer lines, which is not accounted for in the
removal of broadband features in the measured spectrum. This is known as the Ring effect and is thought to be primarily caused





by rotational Raman scattering (RRS) in the atmosphere (Grainger and Ring, 1962; Lampel et al., 2015; Vountas et al., 1998). Correction of the Ring effect is essential for weak absorbers (e.g. BrO) and for when the atmospheric path length is longer, such as measurements early or late in the day, or far from zenith.

The Ring effect is usually corrected for by treating it as another (or sometimes multiple) absorbing gas species. How the Ring spectrum is produced varies. In some cases the inelastic scattering efficiency is assumed to be independent of wavelength, so the Ring spectrum is taken as a normalised inverse Fraunhofer spectrum. A more robust method is to model the Ring effect by applying known RRS efficiencies for atmospheric species to the Fraunhofer spectrum. In either case a base Ring spectrum, $R(\lambda)$, is included in the fitting process using a scaling parameter $r$:

$$\tau'(\lambda) = \sum_i [\sigma_i'(\lambda) \cdot \alpha_i] + (R(\lambda) \cdot r) \tag{9}$$

## 2.3 iFit

### 2.3.1 Forward model

The main aim of iFit was to eliminate the issues raised by the measured reference spectrum used in DOAS. Instead, a high resolution Fraunhofer spectrum forms the starting point for a forward model of atmospheric radiative transfer to recreate the measured spectrum. iFit still utilises the Beer-Lambert law to describe the absorption of radiation in the volcanic plume, but remains in intensity space instead of converting to optical depth. The iFit forward model is:

$$I(\lambda) = ILS \otimes \left( I_0^*(\lambda) \cdot \exp\left[ \sum_i [-\sigma_i(\lambda) \cdot \alpha_i] \right] \cdot P(\lambda) \cdot (R(\lambda) \cdot r) \right) \tag{10}$$

Here all symbols are as defined in section 2.2. Note that the ILS convolution is applied only once at the end of each iteration, not to each gas cross-section in question. This means that all calculations take place on the higher resolution wavelength grid of the initial Fraunhofer spectrum rather than the low resolution spectrometer grid. Although computing on the high-resolution model grid is computationally inefficient, it removes the need to correct for the $I_0$ effect, as well as avoiding multiple convolutions during every iteration of the fitting process.

### 2.3.2 Fitting procedure

The iFit fitting procedure is outlined in Fig. 2. The measurement spectrum is first corrected for the dark current, bias, stray light and flat response, and the desired fitting wavelength window is extracted. A model spectrum is then built on the high resolution model grid, which typically has a spacing of 0.01 nm and a wavelength range covering the chosen fitting window with 2 nm of padding on either side to allow for the wavelength shift. The wavelength shift is a common correction in DOAS-style measurements due to changes in the wavelength calibration of the spectrometer during measurements due to internal





temperature changes or due to inaccuracies in the positioning of the diffraction grating (Platt and Stutz, 2008, section 8.3.3). All reference spectra are then interpolated onto the model grid using cubic spline interpolation. Note that this only needs to be performed a single time; once the reference spectra have been interpolated they can be stored for future use.

The reference spectra used here are a high resolution Fraunhofer spectrum (Chance and Kurucz, 2010), gas absorption

spectra for $SO_2$ at 298 K (Rufus et al., 2009), $NO_2$ at 223 K (Voigt et al., 2002), $O_3$ at 223 K (Gorshelev et al., 2014) and BrO at 298 K (Fleischmann et al., 2004). The spectral resolution of all absorption spectra are less than or equal to 0.01 nm, except for BrO which has a resolution of 0.02 nm. A Ring spectrum produced using the QDOAS software is also used (Danckaert et al., 2017, version 3.2). Fig. 3 shows the Fraunhofer spectrum used in the retrievals presented here. This has been smoothed using a Gaussian function (FWHM = 0.5 nm) and compared to a measured skylight spectrum. The spectra have been offset for

clarity. The differences seen are due to $O_3$ absorption at low wavelengths. A background polynomial of a user defined order (typically $3^{rd}$ or $4^{th}$) is included to account for intensity changes, broadband aerosol scattering and transmission through the telescope and fibre. This allows the iFit procedure to cope with the variety of different lighting situations encountered during measurements of sky spectra. Although the dark current, bias and stray light effects are corrected for where possible, fitting in intensity does allow such steps to be ignored as they can also be included in the background polynomial.

Finally the ILS convolution is applied and the spectrum is interpolated from the high resolution model grid onto the measurement grid. The forward model is then fitted to the measured spectrum using the Levenberg-Marquardt method. This is a non-linear least-squares minimisation where the input variables $x$ (also known as the state vector), are optimised to minimise the sum of the squared residual between the measured spectrum and the model fit, which is a function of $x$.

## 3   Results and Discussion

To test the ability of iFit to accurately retrieve $SO_2$ column densities a series of tests were performed, including fitting of spectra from clear sky, $SO_2$ calibration cells and a volcanic plume. This section explores these results, presenting spectra taken at different locations and at different times of day to demonstrate the robustness of iFit under different lighting conditions. In each case $SO_2$, $NO_2$, $O_3$ and BrO were included in the fit, along with a $3^{rd}$ order background polynomial, a Ring spectrum and a wavelength shift and stretch. The ILS used throughout is a Gaussian function, and any error values given are produced from

the covariance of the fitted parameters. A Gaussian function was chosen as it accurately fits the shape of the measured emission line from a mercury lamp (HG-1 from Ocean Optics).

The wavelength region used for these results (304 - 320 nm) is common to most scattered sunlight retrievals of $SO_2$, although some work has been done on higher wavelength bands such as 360 - 390 nm (Bobrowski et al., 2010). It is used as $SO_2$ shows a strong absorption spectrum below 320 nm, with the lower limit set by the strong absorption edge of $O_3$ which effectively

blocks all solar radiation below 300 nm.





## 3.1 Clear Sky

An important test for iFit is to accurately fit spectra when no plume is present to show that the background sky spectrum can be accurately reproduced. Fig. 4 shows an example of a fitted clear sky spectrum taken at Masaya volcano, Nicaragua at 9:30 am (CST) on 14$^{th}$ January 2018 using a Flame spectrometer (FLMS02101) with an purely Gaussian ILS with a width of 0.56

nm. Frame (c) of Fig. 4 shows the measured and synthetic absorbance spectra for $O_3$. The measured spectrum (blue line) is produced by dividing the measured spectrum by the model fit with the $O_3$ contribution removed and taking the negative natural logarithm. The synthetic spectrum (orange line) is produced using the retrieved $O_3$ column density as an input into equation 1 (with an ILS convolution applied). A similar process is applied in frame (d) for the Ring spectrum. The retrieved $SO_2$ column density is -3 ($\pm$7) ppm.m and the standard deviation of the residual is 0.6%. There is some structure in the fit residual (also

seen in subsequent fits) which remains constant during measurements. This residual could have a number of sources, including the presence of a trace absorbing gas not included in the fit, inaccuracies in the reference spectra used or unaccounted for instrumental effects.

## 3.2 $SO_2$ Cells

### 3.2.1 Stationary measurements

Spectra were acquired using calibration gas cells filled with a known column density of $SO_2$. A total of 9 cells were tested, with column densities of 54, 93, 107, 195, 482, 520, 993, 1027 and 1802 ppm.m (with stated uncertainties of $\pm$ 10%). All measurements were performed from the roof of INGV Catania, Italy (37° 30' 48" N, 15° 4' 56" E), under clear blue sky conditions. The 54, 107, 520 and 1027 ppm.m cell spectra were taken on 7$^{th}$ September 2017 and all others were taken on 10$^{th}$ July 2018. Roughly 20 spectra were taken of each cell using a USB2000+ spectrometer (serial number: USB2+H15972). Fig.

5 shows an example fit of a spectrum taken of the 520 ppm.m calibration cell. The retrieved $SO_2$ column density is 496 ($\pm$11) ppm.m and the standard deviation of the residual is 1.3 %.

The calibration cells were used to test the impact of the spectrometer ILS and fitting window used on the accuracy of the retrieved $SO_2$ column density. Firstly the ILS was tested with values from 0.42 – 0.62 nm in 0.04 nm intervals (Fig. 6). Here the stated ILS width is the Full Width Half Maximum (FWHM) of a purely Gaussian line shape. All retrievals were performed

using a fit window of 305 – 318 nm. From these results it is clear that using the correct ILS is essential to retrieving the correct $SO_2$ column density, as the wrong value can lead to significant systematic under- or overestimations, as well as an incorrect zero value. The ILS was measured using a mercury lamp to be 0.50 ($\pm$0.01) nm (using the 334.15 nm line). Errors given by the iFit retrieval are typically 10 – 25 ppm.m and so are too small to be seen on the figure.

To test the effect of fitting window on retrieved $SO_2$ the same spectra were analysed using different 10 nm wide fitting

windows, with starting wavelengths between 304 – 310 nm (Fig. 7). These results show that the fitting window has a smaller impact on the retrieved $SO_2$ column density, but is still important to consider. In particular, use of higher wavelengths leads to an overestimation in the retrieved $SO_2$, possibly due to the reduced strength of the $SO_2$ absorption spectrum at higher wavelengths. This is important for operational measurements as the amount of light available at lower wavelengths (< 310 nm) can be very





low at long atmospheric path lengths due to the strong $O_3$ absorption edge. This can make measurements taken early or late in the day (e.g. before 10:00 or after 15:00) difficult to analyse using lower windows. Additionally, for very high $SO_2$ column densities, the plume can become effectively opaque within the strongest absorption bands, so the higher wavelengths need to be used. It is important to consider these effects on the choice of fit window. Fitting in intensity rather than optical depth makes it easier for these problems to be identified and corrected for in the field, which is vital for operational monitoring or campaign measurements.

### 3.2.2 Scanning measurements

Here we will compare iFit to the retrieval methodology used by the FLAME scanning network on Etna and Stromboli volcanoes in Italy. This retrieval method is a hybrid between traditional DOAS and iFit: a synthetic sky spectrum is used to convert the measured intensity spectrum to an optical depth spectrum, which is then fitted in the same manner as traditional DOAS (Salerno et al., 2009b). Here we use iFit to analyse the same spectra presented in section 4.2.2 of Salerno et al. (2009b). The test is described in full by the authors, so only a brief description will be provided here. Two $SO_2$ cells were mounted to the rotating scanner head of the "ENIC" station to the south of Etna on days when the volcanic plume direction was away from the station location. Spectra were taken on 2nd and 4th August 2006 using 345 and 130 ppm.m calibration cells respectively. The cells have a reported uncertainty of 5%. The station performs anti-clockwise scans (east to west) covering the sky 12° above each horizon (12° to 168°) in 1.5° increments. Each spectrum was taken with an integration time of 200 ms and averaged across 10 spectra. Each scan took approximately 5 minutes to perform and consists of 105 measurement spectra and a single dark spectrum. Upon inspection of the data the last spectrum of each scan was found to have a very low intensity, possibly due to an obstruction to the line of sight of the scanning head. For this reason the last spectrum of each scan has been ignored. The results are shown in Fig. 8.

For both cells there is a systematic over- or underestimation depending on the viewing direction of the scanner. The retrieval of the 345 ppm.m cell shows a systematic underestimation from the stated cell amount. The spectra were originally analysed with the current technique used by the FLAME network as described by Salerno et al. (2009b). The iFit retrieval was compared to these data (Fig. 9). The average and standard deviations for both methods across the full time series are given in table 1. iFit gives nearly identical results to the FLAME analysis, but with less overall scatter. Both methods retrieve a lower value for the 345 ppm.m cell, suggesting that the actual cell $SO_2$ amount is lower than stated. The average error in the iFit retrieval is of the order of the standard deviation, showing that the uncertainty arising from viewing angle is comparable to that in the fit retrieval.

It is important to note that it would be impossible to perform a traditional DOAS retrieval on this dataset due to the need for frequently updated clear sky spectra. This highlights one of the main advantages of using a synthetic sky spectrum in retrievals. We would also like to emphasise that iFit offers an improvement over the FLAME method as the FLAME method uses a number of different ILS convolutions applied at different stages to fit the measured spectrum, making it a non-physical description of the situation. iFit only applies one ILS at the end of the model (upon the light entering the spectrometer) and so remains a physical description of the radiative transfer processes at work.



### 3.3 Volcanic Plume

Here we present results from car traverses of the plume from Masaya volcano in Nicaragua on 14[th] January 2018. Fig. 10 shows an example plume spectrum and fit, along with the $O_3$ and $SO_2$ absorbance spectra. The retrieved $SO_2$ column density is 398 ($\pm$8) ppm.m and the standard deviation of the residual is 0.5%.

These spectra were also analysed using a basic DOAS retrieval as described in section 2.2, including a fitted $SO_2$ spectrum, $O_3$ spectrum, Ring spectrum and a wavelength shift and stretch. The $SO_2$ cross section was corrected for the $I_0$ effect using a column density of 400 ppm.m. The $SO_2$ time series for two traverses are shown in Fig. 11. The reference spectrum for the DOAS retrieval was taken between the two traverses at approximately 9:59. Both retrievals were performed using a wavelength window of 305 – 318 nm and with an ILS width of 0.56 nm. The $SO_2$ column retrieved using iFit agrees very well with those

of DOAS.

### 3.4 Computational Speed

Performing the fit on the high resolution model grid has the benefit of avoiding issues such as the $I_0$ effect and multiple ILS convolutions – however it is more computationally costly than using the lower resolution spectrometer grid. This means that the iFit procedure shown here could be ill-suited to analysis of large datasets (e.g. long time series of satellite data). Using

the case of the Masaya traverses shown in section 3.3 a simple timing test was performed to assess the speed of the iFit procedure. This test was performed on a desktop computer (HP EliteDesk with an Intel i5 processor, 3.20 GHz) running the iFit software (written in Python 3.6). Analysis of the 161 spectra took 32.1 seconds (0.20 seconds/spectrum) which is smaller than typical sampling periods used for volcano monitoring (approximately 1 second). The program speed can be increased by using the results of the previous fit to inform the first guess of the next, in which case the analysis takes 14.3 seconds

(0.09 seconds/spectrum). As a comparison the analysis using the DOAS code took 24.3 seconds (0.15 seconds/spectrum). This shows that iFit could be used to analyse spectra in real time, making it well suited to both campaign-style measurements and operational volcano monitoring.

### 4 Conclusions

We have shown that it is possible to accurately recreate measured UV spectra using a forward model built on a high resolution

synthetic reference spectrum. The main advantage of this over the traditional DOAS methodology is that there is no requirement for a clear sky reference spectrum. This reduces the complexity of performing measurements and lowers the possibility of introducing systematic errors from $SO_2$ contaminated reference spectra. Use of a high resolution forward model also removes the need for correction of the $I_0$ effect. We have also shown that iFit is sufficiently fast to analyse spectra in real time.

Fitting in intensity can be advantageous over optical depth as it minimises the number of transformations applied to the

measured spectrum, further reducing the potential for inclusion of systematic errors into the retrieved $SO_2$ column density. Remaining in intensity also reduces the impact of the dark current, bias and stray light on the fit as these can be bundled into



the background polynomial, which is a significant advantage over fitting in optical depth. Finally fitting in intensity is more intuitive than fitting in optical depth. While this is not important mathematically, it does mean that potential issues such as low intensity values of lower wavelength light can be identified easily and corrected for during measurements. This can be important for operational volcano monitoring or campaign style measurements, especially for non-experts performing these

measurements.

We have shown that iFit is able to accurately fit spectra from the clear sky, calibration cells and a volcanic plume. The effect of ILS width and wavelength fitting window have been investigated using $SO_2$ cell data. The ILS width has a higher impact on the retrieved $SO_2$ column density than the choice of wavelength window and must be characterised accurately. Additional care must be taken if the spectrometers are not temperature stabilised as this can lead to changes in the ILS between or during

measurements.

iFit was compared to current retrieval methods - firstly to the hybrid method currently used by the FLAME scanning network on Etna and Stromboli in Italy, and secondly to traditional DOAS retrieval for a car traverse of the plume of Masaya volcano in Nicaragua. iFit agrees within uncertainty with each method, suggesting that it is robust and can be routinely used in volcano monitoring. The lack of a requirement for a reference spectrum means that iFit would be especially well suited to deployment

in permanent scanning stations. iFit could also improve automated traverse measurements, for example using public transport such as those described by (Mori et al., 2017), as iFit would allow for real time automated analysis of the collected spectra.

In the future iFit could be extended to correct for the effects of light dilution (Kern et al., 2012; Mori et al., 2006). iFit will allow for a more direct treatment of light dilution as the background sky and plume spectra can be separated explicitly in the forward model. This, as well as retrievals of other volcanic gases such as BrO and OClO, is the subject of ongoing work.

*Code and data availability.*  The code and spectra presented here are available at https://github.com/benjaminesse/iFit

*Author contributions.*  B.E. designed the study, developed the analysis code, performed all iFit analyses and drafted the manuscript. G.S. provided FLAME network data. M.B., M.V. and R.K. aided in the development of the model and software. All authors reviewed the manuscript.

*Competing interests.*  The authors declare no competing interests.

*Acknowledgements.*  We thank the staff of INETER in Nicaragua, in particular Marta Ibarra Carcache and Juan Lopez, for their vital as-
sistance during measurements at Masaya volcano. This work was conducted during a PhD study supported by the Natural Environment Research Council (NERC) EAO Doctoral Training Partnership, and is funded by NERC whose support is gratefully acknowledged [grant number NE/L002469/1].





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



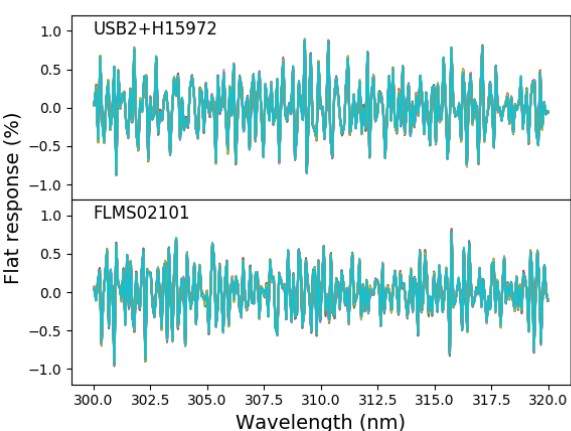

**Figure 1.** Example flat spectra for two Ocean Optics spectrometers: a USB2000+ (USB2+H15972) and a Flame-S (FLMS02101). 10 flat spectra for each spectrometer are shown. Each spectrum is the average of 1000 spectra, each with an integration time of 30 ms. The spectrum is normalised with a boxcar smoothed spectrum. These measurements were performed with an Ocean Optics DT-MINI-2-GS light source.



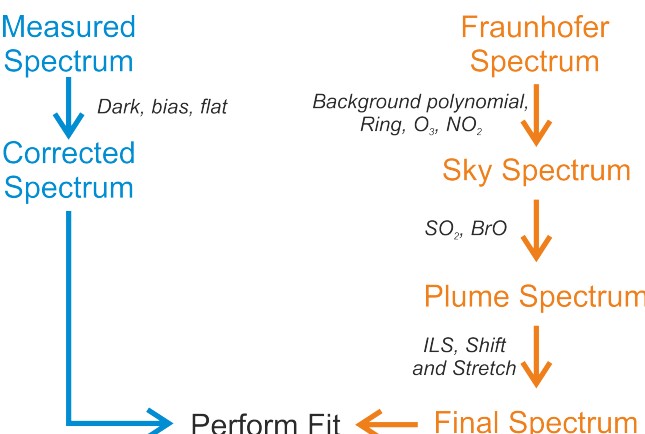

**Figure 2.** Block diagram for the iFit fitting procedure. The left-hand blue column corresponds to the measured spectrum, while the right-hand, orange column corresponds to the forward model.




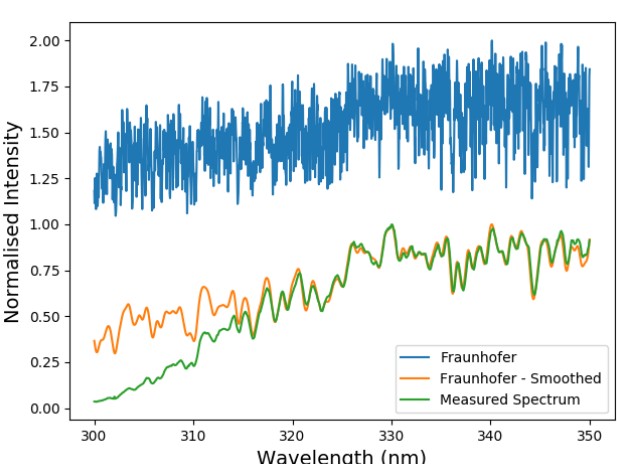

**Figure 3.** The high resolution Fraunhofer spectrum used in iFit. This has been smoothed using a Gaussian function (FWHM = 0.5 nm) and plotted with a measured skylight spectrum for comparison. All spectra have been normalised for clarity.



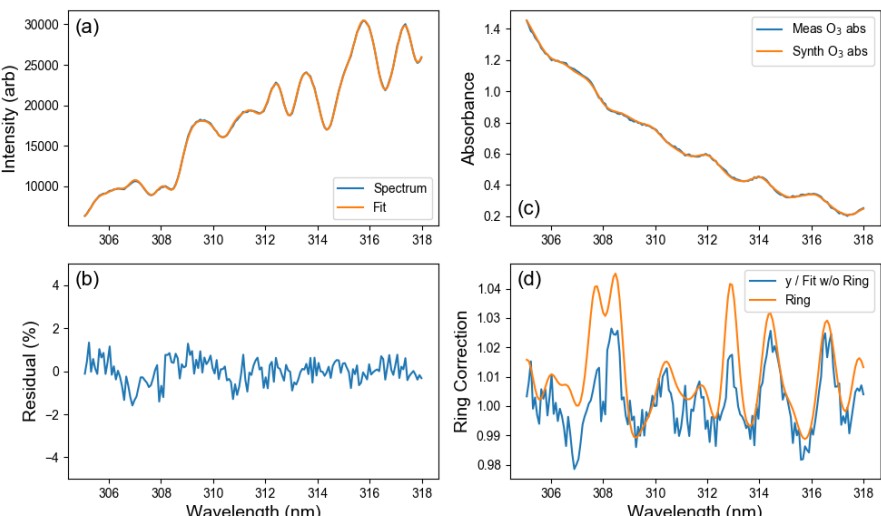

**Figure 4.** Example fit for a clear sky spectrum with the FLMS02101 spectrometer. (a) The measured spectrum (blue line) and model fit (orange line) cut to the desired fit window. (b) Percentage fit residual. (c) Measured and synthetic absorbance spectra for $O_3$. (d) Measured and synthetic Ring spectra.




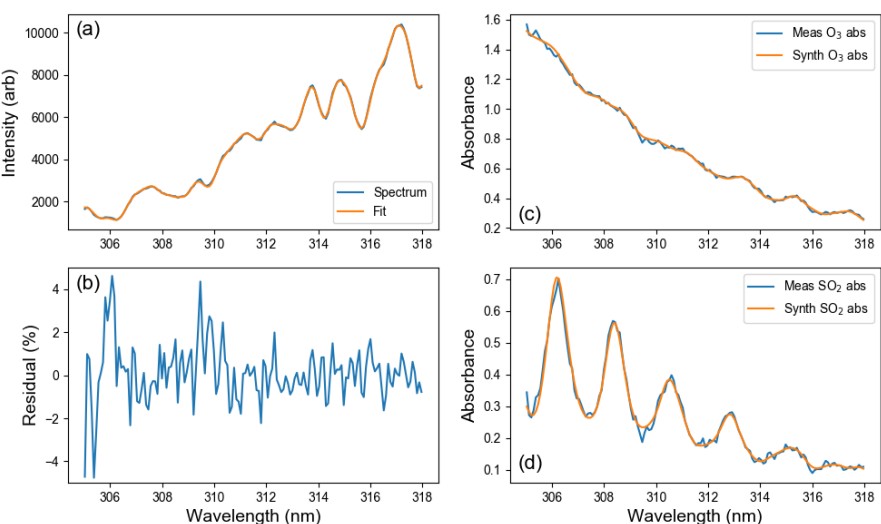

**Figure 5.** Example fit for a 520 ppm.m cell spectrum with the USB2+H15972 spectrometer. (a) The measured spectrum (blue line) and model fit (orange line) cut to the desired fit window. (b) Percentage fit residual. (c) Measured and synthetic absorbance spectrum for $O_3$. (d) Measured and synthetic absorbance spectrum for $SO_2$.





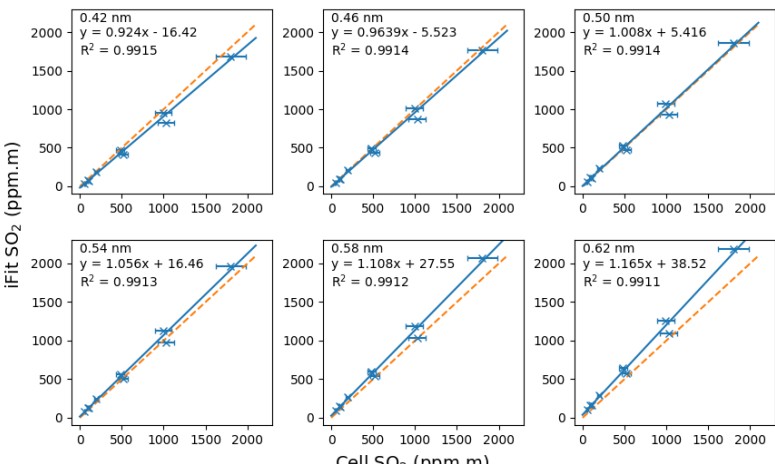

**Figure 6.** Retrieved SO$_2$ column densities against stated cell amounts for different ILS widths. The dashed orange line is the line y = x, the solid blue line is the fitted linear regression of the retrieved cell amounts. The ILS width, fitted gradient, y-intercept and R$^2$ values are given in the captions. All spectra were fitted between 305 – 318 nm.





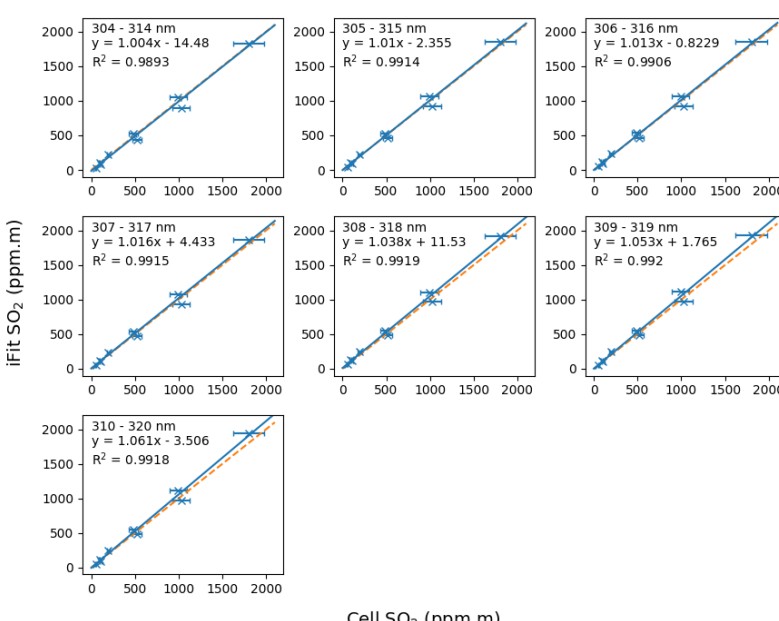

**Figure 7.** Retrieved $SO_2$ column densities against stated cell amounts for different fitting wavelength windows. The dashed orange line is the line y = x, the solid blue line is the fitted linear regression line of the retrieved cell amounts. The wavelength window, fitted gradient (m), y-intercept (c) and $R^2$ values are given in the captions. All spectra were fitted using a 0.50 nm ILS width.




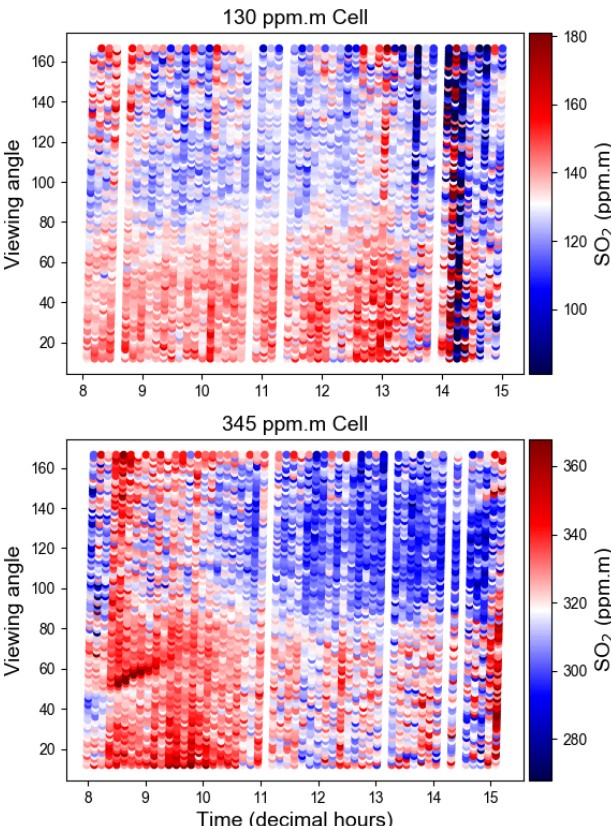

**Figure 8.** Retrieved SO$_2$ column densities against viewing angle and time for the stationary scanner tests. Red and blue colours represent higher and lower values compared to the average respectively. An ILS width of 0.6 nm and fitting window of 305 – 318 nm was used. Gaps in the data are periods of very low intensity spectra that resulted in poor fits. These have been manually removed from the dataset to improve clarity.



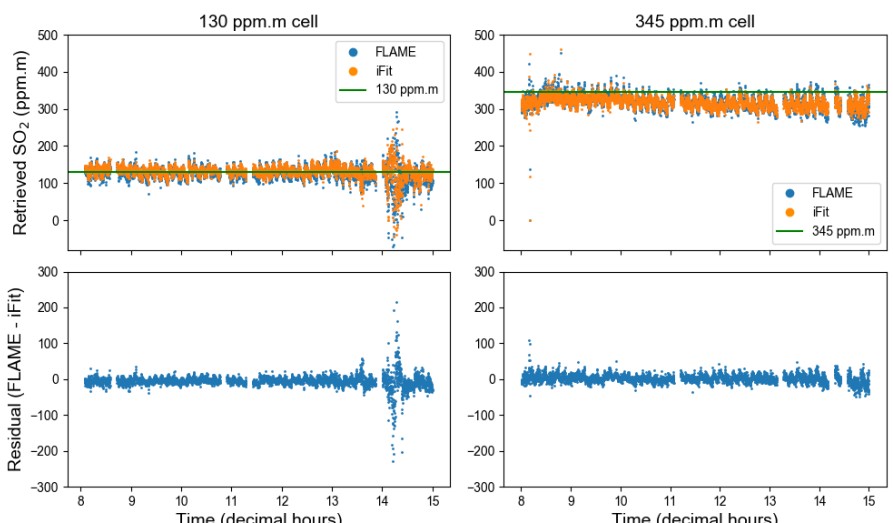

**Figure 9.** Retrieved SO₂ column densities for cell spectra taken using the ENIC station of the FLAME network on Etna. The results of both the FLAME analysis algorithm (blue circles) and iFit analysis (orange circles) are shown, as well as the residual between the two. The same spectrometer ILS width of 0.6 nm was used for both methods. Gaps in the data are periods of very low intensity spectra that resulted in poor fits. These have been manually removed from the dataset to improve clarity.





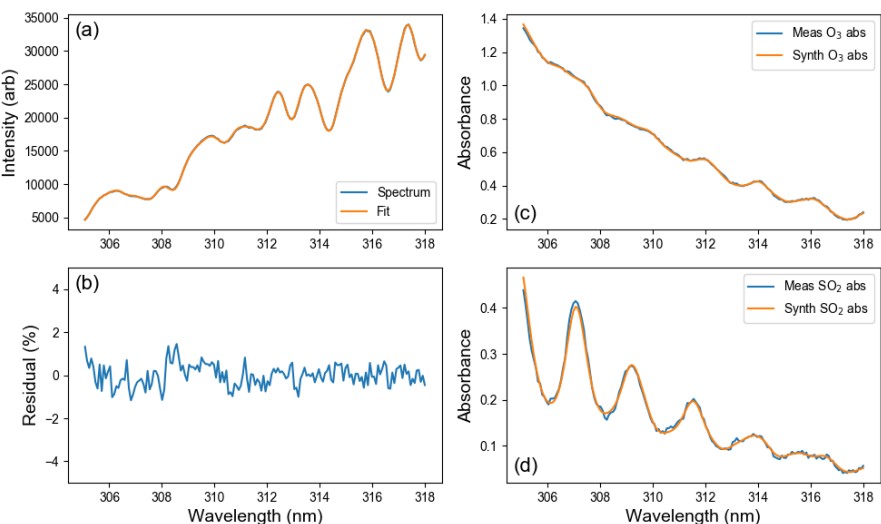

**Figure 10.** Example fit for an SO$_2$ rich plume spectrum taken with the FLMS02101 spectrometer during a car traverse. (a) The measured spectrum (blue line) and model fit (orange line) cut to the desired fit window. (b) Percentage fit residual. (c) Measured and synthetic absorbance spectrum for O$_3$. (d) Measured and synthetic absorbance spectrum for SO$_2$.



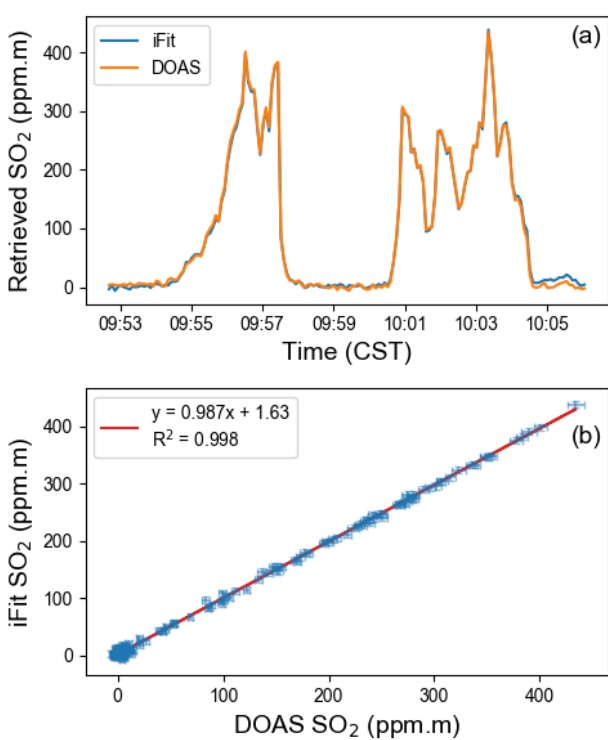

**Figure 11.** Two traverses of the plume from Masaya volcano. (a) Retrieved $SO_2$ column densities for two car traverses of the plume of Masaya Volcano, Nicaragua using iFit (blue line) and DOAS (orange line). (b) Scatter plot of the iFit and DOAS retrieval for each spectrum. Errors are produced from the covariance of the fitted parameters in the least-squares minimisation.



**Table 1.** A summary of the retrieved SO$_2$ column densities from the scanning measurements for both iFit and the FLAME analyses. The uncertainty on the average value is the average error given by the fitting process.

|  | iFit | | FLAME | |
| --- | --- | --- | --- | --- |
|  | Average | Standard Deviation | Average | Standard Deviation |
| 130 ppm.m | 131 (±16) | 17 | 126 (±17) | 22 |
| 345 ppm.m | 318 (±15) | 16 | 320 (±18) | 22 |