# Peer review of "iFit: An intensity based retrieval for volcanic SO2 from scattered sunlight UV spectra"

_Atmospheric Measurement Techniques, 2018_

## Referee Comment (RC1) · Anonymous Referee #1 · 2 Mar 2019

The manuscript "iFit: An intensity based retrieval for volcanic SO2 from scattered sunlight UV spectra" by B. Esse and co-authors present an alternative method for the evaluation of SO2 column densities from of UV/vis spectra. In contrast to classical DOAS, their method does not require a measured reference spectrum, which may display a crucial limitation for measurements at volcanoes when the plume contaminates the entire field of view of the instrument.

The topic is suited to be published in AMT. The manuscript, however, requires major revisions as detailed below. The overall goal should be that the method is formulated as clearly as possible to allow fellow scientists to reproduce the results.

* Major comments *

[Figure]

1) In the Abstract (l. 5) the authors claim to present a novel technique using intensity fitting and a solar reference. This claim, in my opinion, is not fulfilled because, in general, intensity fitting is equivalent to fitting in optical density space and commonly applied in the IR community. Secondly, using a literature solar reference was already applied by Salerno et al., 2009b and discussed more thoroughly by Lübke et al., 2016. What makes the described approach interesting is the calculation in high resolution. This detail, however, is not motivated enough in the abstract/manuscript. Why should it be advantageous to fit in high resolution while the actual measurement spectrum does not provide any information in this increased resolution anyway?

I would like to take this opportunity to mention that there are numerous publications addressing the correction of simplifications introduced by DOAS. These were compiled by Referee #1 in his/her comment on Burton and Sawyer, 2016. The discussion from the Referee comment should be included in the introduction or the discussion of the manuscript.

2) The method section states that multiplications and convolutions were non-commutative (p. 4, l. 29). This is wrong, because a convolution is an integral, which is linear (see https://en.wikipedia.org/wiki/Integral). Therefore, all conclusions drawn from this hypothetical non-commutitativeness are also wrong. In DOAS, the convolution is therefore not required to be applied in every fit iteration (p. 5, l. 1). Please revise the manuscript an remove implications of this erroneous assumption.

3) The forward model (Eq. (10)) approximates broadband absorption with a polynomial factor $P(\lambda)$ in intensity space. However, Eq. (3) suggests that it should be rather $\exp(P(\lambda))$. What is the influence of this approximation on the fit result?

4) In my opinion, the formulation of the inversion problem is not completely clear and should be improved (Sect. 2.3.2). - Please detail, which parameters are fitted (or included in the state vector), eventually in an additional table. Please also include the initial values. - How are stretch and shift treated? They do not appear in Eq. (10).

[Figure]

- Is the fit applied on filtered spectra as stated in p. 4, l. 18f? - In my opinion, a real advantage of fitting in high resolution is the opportunity of retrieving an actual ILS parametrisation simultaneously. Has this option be considered? If not, why? - Figure 2 is a bit confusing. It seems that the "Final Spectrum" is entirely calculated before fitting. Please specify, which steps are done before fitting and which parameters are included in the fit. The fit is usually an iterative process, which could be sketched in the figure. - What are the termination criteria of the fit procedure?

5) The iFit results are compared to classical DOAS results in Sect. 3.3. The description of the applied DOAS method, for which the reader is referenced to Sect. 2.2 (p. 10, l. 5), is not complete. For instance, it is not clear, which software was used. From the provided description one may conclude that did not apply a commonly used and validated implementation like QDOAS. In my opinion, a standard software package should be used in order to provide a meaningful comparison between iFit and DOAS.

6) Bobrowski et al. 2010 state a "standard evaluation range of approximately 310 to 325nm". What is the effect of shifting the range for the DOAS retrieval to 305-318 as applied in this study? I would assume that interferences from O3 absorption and Rayleigh scattering are more dominating at shorter wavelengths. Please discuss this issue in the revised manuscript.

7) The results of the comparison between iFit and DOAS are compiled in Fig. 11. The discussion of the comparison, however, is a bit meagre. For example, Fig. 11 reveals a bias compared to the background (and the DOAS) value at the left and right tail. The reference was taken at 9:59, which is approximately in the centre of Fig. 11 (a). At 10:05, however, iFit results seem to be significantly positively biased. Please discuss this issue in the manuscript. Furthermore, a quantitative comparison between iFit and DOAS like compiled in Table 1 would be favourable.

8) When treating the I0 effect and strong absorption, the knowledge about the ILS is crucial. It seems that a Gaussian ILS was applied throughout the study (p. 7, l. 27). The

paper by Beilre et al., 2017 investigates parametrisations of different spectrometers and states, that the Gaussian may be a sufficient ILS parametrisation only for some instruments. Therefore, I suggest to add a plot showing the measured ILS for the applied instruments.

9) I am a bit confused about the discussion of the computational speed (Sect. 3.4). Actually, iFit requires a convolution in every fit step and, therefore, an inferior performance of iFit compared to DOAS can be anticipated a priori.

As a comparison, Beirle et al. (2013) claim to be able to process non-linear DOAS analysis using 0.004 seconds per fit with DOASIS. When they implemented their linearised method in MATLAB, they even achieved 0.00004 seconds per fit. In the study here, a faster CPU with 3.2 GHz (instead of 2.5 GHz) was applied and achieved not less than 0.09 seconds per spectrum. Hence, iFit is more than a magnitude slower than DOASIS on a slower PC.

Furthermore, the stated observation that changing the initial value could speed up the process by more than a factor of 2 indicates that either the algorithms converges very slowly or that the standard initial values are not chosen optimally. Please discuss.

In my opinion, computational speed should not be overrated for the application of a scientific algorithm. If the scientific question requires a slower data evaluation algorithm, it shall be favoured over a faster and less accurate one. The algorithm presented here works without a measured reference spectrum, which is a quality of its own. Therefore, I suggest to remove the discussion about computational speed from the manuscript altogether (also: p. 10, l. 28 "We have...")

10) I do not agree with the authors statement that fitting in intensity space is more intuitive (p. 11, l. 1). This is a personal opinion and should be omitted. For me, absorption is an asymptotic process, which is intuitively linearised by transforming from intensity space to optical density space.

[Figure]

* Minor comments *

- I am a bit confused about the identity of the first author. Is his first name Ben or Benjamin? Personally, I would refrain from using nicknames in scientific author lists and affiliations.

- p. 1, l. 8: "number of advantages" -> "primarily" is a bit vague. Please be more specific.

- p. 1, l. 10: remove the repeating "without the use of a reference spectrum"

- Please avoid qualitative statements like p. 1, l. 9: "accurately" p. 1, l. 10: "well suited" p. 1, l. 11: "strong potential" p. 7, l. 20: "accurately" p. 10, l. 9: "very well" p. 10, l. 24: "accurately" p. 11, l. 6: "accurately" and use quantitative statements instead.

- Sect 2.2.2 Please add a statement that the I0 effect is due to the non-commuting of exponential function and convolution. This effect can be corrected for in DOAS evaluations.

- Eq. (8): The clarity of this formula could be improved, e.g. by omitting "$(\lambda)$" and some more brackets. Maybe also use \frac{}{} as in Eq. (A3) in the cited Aliwell et al. 2002.

- p. 5, l. 26 omit "another"

- p. 10, l. 31: Please remove the sentence starting with "Remaining ..." because it is not based on findings in the paper. All effects mentioned can be addressed by DOAS.

- p. 11, l. 3: This is not a particular improvement of iFit, because low intensity issues are visible in raw spectra. Please omit this statement.

- Figure 4(d): Caption and legend are not matching. What is the difference between measured and synthetic Ring? Please clarify.

- Caption of Fig. 11: Please detail the definition of CST time.

- Please add gridlines to all plots to improve readability.

- Please add subplot labels like (a), (b) and so forth to Figs. 1, 6, 7, 8, and 9 for better reference.

- Please arrange the subplots in Figs. 4, 5, and 10 first left to right then top to bottom.

* References *

Beirle, S., Sihler, H., and Wagner, T.: Linearisation of the effects of spectral shift and stretch in DOAS analysis, Atmos. Meas. Tech., 6, 661-675, https://doi.org/10.5194/amt-6-661-2013, 2013.

Beirle, S., Lampel, J., Lerot, C., Sihler, H., and Wagner, T.: Parameterizing the instrumental spectral response function and its changes by a super-Gaussian and its derivatives, Atmos. Meas. Tech., 10, 581-598, https://doi.org/10.5194/amt-10-581-2017, 2017.

Burton, M. R. and Sawyer, G. M.: iFit: An intensity-based retrieval for SO2 and BrO from scattered sunlight ultraviolet volcanic plume absorption spectra, Atmos. Meas. Tech. Discuss., https://doi.org/10.5194/amt-2015-380, in review, 2016.

---

## Referee Comment (RC2) · Anonymous Referee #2 · 5 Mar 2019

Remote sensing of volcanic SO$_2$ emissions via Differential Optical Absorption Spectroscopy (DOAS) has become a major tool in volcanology. DOAS analyses differences between a measured light spectrum and a background spectrum. The background spectrum is usually recorded by the same instrument in the temporal proximity to the measured spectrum but in another viewing direction. Using such a background spectrum allows DOAS for an automatic correction of most instrumental effects, stratospheric effects, and shared tropospheric effects. As a drawback, this method is insensitive for a contamination of the background with e.g. volcanic SO$_2$ and DOAS then potentially underestimates the absolute volcanic SO$_2$ emissions. Salerno et al. (2009) and Lübcke et al. (2016) presented alternative approaches which use a synthetic background spectrum in order to detect/circumvent a possible background contamination.

[Figure]

Esse et al. present another approach *iFit* using a synthetic background spectrum. Advances beyond the state of the art are desirable and AMT is in principle a suitable journal for such a topic. It is however not clear to me whether their proposed approach poses a substantial advance. In particular, it is not possible for me to assess their approach (1) because the approach is not described in sufficient detail and (2) because its performance is not set in contrast to the state of the art. In addition, some wrong or ambiguous statements on DOAS raises concerns whether Esse et al. apply DOAS in the best available way.

In conclusion, I can not support the publication of the manuscript in the presented form. For publication, *iFit* has to be described - first of all and at the very least - in such a way that the reader is able to reproduce their results. Second, I highly recommend a quantitative and more comprehensive comparison of *iFit* with the approach from Lübcke et al. (2016) and with the standard DOAS approach in order to provide evidence that *iFit* in fact provides an advance beyond the state of the art. Thirdly, I recommend to neglect all redundant and subjective statements from the manuscript in order obtain maximum clearness. Please find below a detailed list of the major objections.

Finally, I expect that tackling these major objections would result in massive changes of most parts of the manuscript. In consequence, a second subsequent review appears to be mandatory which focusses on the then provided spectroscopic details of their approach.

**1   No comprehensive description of their method is provided**

I am aware that Esse et al. uploaded their used data and code written in python. The proposed approach has to be reproducible in principle with any programming language/platform/python version. Therefore, this review does not assess the python code but exclusively assesses the provided manuscript.

Esse et al. sketch the architecture of their approach in Figure 2, however, mathematical descriptions or measurement instructions are not provided for each step. Therefore, iFit can not be reproduced by the reader. In particular, the following steps have to be provided/clarified:

1. Page 6, lines 27-29: *"A model spectrum is then built on the high resolution model grid, which typically has a spacing of 0.01 nm..."* How is this model spectrum build? By a convolution with a Gaussian ILS with a FWHM of 0.01 nm?

2. How are the effects of the dark current and "bias" determined and corrected? Remark: the latter is typically called "offset" rather than "bias" (see e.g. Platt and Stutz, 2008).

3. Flat spectrum: Esse et al. retrieved the "flat spectrum" by a simple averaging. In contrast, Lübcke et al. (2016) retrieved the instrument effects (flat spectrum but also further instrument effects such as temperature effects) by a Principal Component Analysis (PCA). For me, the PCA approach appears to be more comprehensive. See also my comment 2.3. Please motivate the choice of a simple averaging instead.

4. Background polynomial: Is equation (10) correct? Physically, all light attenuation effects are on the same footing, i.e. both absorption and scattering effects are summands in the argument of the exponential function. In principle, the scattering effects can of course be written in the presented way e.g. as $P(\lambda) = \exp(P^*(\lambda))$ where $P^*(\lambda)$ is the "real" broad-band scattering polynomial. But this is strictly different from the polynomial as it is used in DOAS. In particular, its coefficients will be different to the polynomial $P(\lambda)$ denoted in equation (3). Please clarify.

5. Ring spectrum: First, I have analogous doubts concerning $R(\lambda)$ in equation (10) and equation (9). Second, please make more explicit how is the Ring spectrum retrieved.

6. Sky spectrum: How is it constructed? By adding the absorption effects of background $O_3$ and $NO_2$? If yes, how is the correct background amount determined?

7. Plume spectrum: Is also $O_3$ included in the fit step from sky spectrum to plume spectrum? If not, how are the diurnal variations of stratospheric $O_3$ contributions and assessed and corrected? Furthermore, why is particularly BrO (but no other gases) included in the $SO_2$ fit scenario? The BrO absorption is rather negligible in the typical $SO_2$ fit ranges.

8. Instrument line shape function ILS: *"The ILS was measured using a mercury lamp to be* $0.50\,(\pm0.01)\,nm"$ (page 8, line 27). The uploaded data does provide ILS (only) at 302 nm (instrument H15972) with a FWHM of 0.58 nm and at 301 nm (instrument FLMS02101) with a FWHM of 0.60 nm. Please provide the full mercury spectra for a presented instruments. (Remark: both uploaded ILS have indeed an about Gaussian shape. A super-Gaussian model proposes exponents of (only) 2.3 and 2.1 and the asymmetry is rather small as well.)
The ILS is a unique property of the instrument, although it varies in general with temperature and wavelength. Accordingly, for a given instrument (and similar temperature) all convolution operations have to use one identical ILS. Applying different ILS can cause a significant decrease in accuracy. Furthermore, all compared spectroscopic should apply an ILS retrieved at the same wavelength in order to be consistent. Ideally, this wavelength is chosen in the wavelength range, e.g. at 315 nm. For practical reasons, the mercury line either at 302 nm or at 334 nm should be chosen. Please clarify which measurement results for the ILS are used. I propose to add a table to the manuscript which lists all instruments used in this study and their spectroscopic properties.
However, later a modelled ILS with *"an ILS width of 0.56 nm"* (page 10, line 9) has been used instead of the measured ILS. Please clarify why instead of the exact measurement results an apparently wrong ILS is used at this step.

9. How are the wavelength shift and stretch determined and corrected?

10. Furthermore, physical-logical the wavelength shift and stretch should be actually applied on the measured spectrum in order to correct for temperature-driven variations of the instrument during the measurement. Analogously, the "flat spectrum" should be applied on the simulated spectrum in order to correct for the instrument effects of the real instrument. I can imagine that these inconsistencies are mathematically identical and may thus lead to the same results. Please clarify why/whether these inconsistencies are required.

11. What actually does "perform fit"? Is it an ordinary DOAS fit? Or is it iteratively minimising $\tau = \log(\frac{I_{right\,hand\,side}}{I_{left\,hand\,side}})$ by means of varying the SO$_2$ column density?

12. Is a stray light correction applied?

**2  Missing comparison with the state of the art**

A comparison with the state of the art is required to provided evidence that *iFit* adds substantial value to the literature. Thereby, it should made clear under which scenarios *iFit* may improve the state of the art and under which scenarios it does not.

1. The manuscript is motivated by the possible underestimation of the SO$_2$ slant column density in a volcanic gas plume. However, no *iFit* results for such a scenario have been provided. Please explain this inconsistency.

2. According to the list of literature provided in the manuscript (and to my knowledge), the approach from Lübcke et al. (2016) is the current state of the art to face such background contamination issue. I highly recommend a direct quantitative comparison of these two methods when applied on the same data (ideally contaminated data) in order to reveal the major differences.

3. Esse et al. propose to use *iFit* for evaluating data recorded by permanent monitoring stations. Monitoring stations are typically not temperature controlled in order to improve their robustness and to lower their power consumption (see e.g. Galle et al., 2010). *iFit* has been tested exclusively for temperature stabilised instruments and Esse et al. concluded that *"care must be taken if the spectrometers are not temperature stabilised"* (page 11, line 9). Accordingly, Esse et al. have not provided evidence that *iFit* is suitable for monitoring stations. This is in particular in sharp contrast to the approach from Lübcke et al. (2016) which presented their results for contaminated data from monitoring stations.

4. I agree with Esse et al. that also a comparison with standard DOAS (recorded background spectrum) appears to be mandatory. For the arguments stated in the very first paragraph, I expect that a standard DOAS approach performs in general (i.e. for non-contaminated scenarios) better than *iFit*. In contrast, the narrative in the current manuscript is rather one-sided, highlighting possible problems in the DOAS approach only. I highly recommend that any subjective valuations are neglected from the technical manuscript parts (e.g. "methods", and "results"). Differences between *iFit* and DOAS should be discussed later in the "discussion" part where all evaluating statements should be supported by (quantitative) evidence.

5. Esse et al. conclude *"the lack of a requirement for a reference spectrum means that iFit would be especially well suited to deployment in permanent scanning stations"* (page 11, line 14). Permanent scanning stations scan typically from horizon to horizon and thus automatically recorded the reference spectrum. Applying *iFit* thus does not provide any gain in measurement time *in particular* at permanent scanning stations. Furthermore, probably only few permanent measurement stations are at all affected by background $SO_2$ contamination. Accordingly, possible benefits from *iFit* are limited to those stations. Please limit your conclusions with respect to those scenarios where you can provide evidence that

[Figure]

*iFit* at least does not perform more poorly than the alternative approaches.

**3  Curious statements on DOAS**

Several strictly wrong or curious/ambiguous statements create some doubts whether Esse et al. apply DOAS on the state of the art level. Namely:

1. Page 4, lines 11-19: Is there any need to discuss the option of a high-pass filter? Is a high pass filter used in *iFit* or for the DOAS retrieval in this manuscript? If not, this paragraph appears to be redundant. The figures 4, 5, 10 show results of the *total* absorption cross section.

2. Page 4, line 26: *"The ILS can either be a mathematical function (such as a Gaussian)..."* The ILS is a property of the instrument and has in general an arbitrary shape. Although it can be indeed often *approximate in good agreement* by a Gaussian line shape function, the real ILS itself is not a mathematical function!

3. Page 4, lines 28-31: Equation 6 is not true! The convolution operation and the scalar multiplication operation are commutative!

4. Page 6, lines 29-30: *"A wavelength-shift is a common correction in DOAS..."*. This is correct, however, when the spectra are wavelength-calibrated prior to the DOAS fit this shift is typically in the order of $\pm 0.001$ nm. Do Esse et al. refer to the additional wavelength shift parameter which is usually allowed between the measurement spectrum and the absorption cross-sections in order to partially compensate for the convolution with a (slightly) wrong ILS? Anyway, this wavelength shift is typically limited to $\pm 0.2$ nm rather than $\pm 2$ nm. A wavelength shift of 2 nm appears to be absurdly large. Please clarify.
5. Page 7, line 27: *"The wavelength region used for these results (304 - 320 nm) is common to most scattered sunlight retrievals of $SO_2$."* This is not true. In DOAS - the predominant remote sensing technique for volcanic $SO_2$ - the used wavelength range starts almost exclusively at 310 nm (e.g. Lübcke et al., 2016), 312 nm (e.g. Theys et al., 2017; Kern and Lyson, 2018), 314 nm (e.g. Lübcke et al., 2014; Dinger et al., 2018), or 326 nm (e.g. Hörmann et al., 2013). The reason for these lower limits is, that the applied approximation in DOAS are only justified as long as the "absorbance" is not much above 0.1. For the data presented in Figure 10, this means the DOAS retrieval should start not lower than at 314 nm. This limitation does not have to hold for an intensity based fit, however, Esse et al. have to make sure that all presented DOAS data are retrieved for an absorbance below 0.1. Otherwise their DOAS results would be too low and a quantitative comparison between *iFit* and DOAS therefore flawed.

6. Page 8, line 31: *"In particular, use of higher wavelengths leads to an overestimation in the retrieved $SO_2$, possibly due to the reduced strength of the $SO_2$ absorption spectrum at higher wavelengths"*. This interpretation of the findings is not supported by further evidence. Furthermore, the *"reduced strength of $SO_2$ absorption"* can be expected to result in a larger fit error but there is no obvious reason why this should cause a less accurate result. In fact, I would interpret the findings other way round: the lower the wavelength the larger is the underestimation in $SO_2$ due to saturation effects and a decreasing solar background radiation (see Platt and Stutz, 2008). At least for DOAS the "absorbance" should be below 0.1 in order to keep the applied approximations justified. With these fundamental limitations in mind, I consider the results for the wavelength range from 310-320 nm the most accurate. Furthermore, I expect that starting at 314 nm or 326 nm would give larger and even more accurate results in particular for the cells above 500 ppmm. In consequence, *iFit* would overestimate the $SO_2$ slant column density. Please provide evidence for your interpretation. In particular, I

highly recommend to present (additionally or exclusively) DOAS results when the wavelength range starts at least at 310 nm.

**4 Some minor formal objections**

- Inconsistent use of brackets (e.g. compare equation 7b and equation 8)

- The (slant) column densities are sometimes denoted by $a_i$ and sometimes by $\alpha$. They have to be denoted by the same consistent letter throughout the manuscript. Furthermore, they should always hold the index.

- $I_0$ and $I_0^*$ appears in several forms throughout the manuscript. They should be consistently denoted by a strictly constant sign.

**Additional references**

Dinger, F., Bobrowski, N., Warnach, S., Bredemeyer, S., Hidalgo, S., Arellano, S., Galle, B., Platt, U., and Wagner, T.: Periodicity in the BrO/SO2 molar ratios in the volcanic gas plume of Cotopaxi and its correlation with the Earth tides during the eruption in 2015, Solid Earth, 9, 247–266, doi:10.5194/se-9-247-2018, 2018.

Galle, B., Johansson, M., Rivera, C., Zhang, Y., Kihlman, M., Kern, C., Lehmann, T., Platt, U., Arellano, S., and Hidalgo, S.: Network for Observation of Volcanic and Atmospheric Change (NOVAC)—A global network for volcanic gas monitoring: Network layout and instrument description, Journal of Geophysical Research: Atmospheres, 115, doi:10.1029/2009JD011823, 2010.
Hörmann, C., Sihler, H., Bobrowski, N., Beirle, S., Penning de Vries, M., Platt, U., and Wagner, T.: Systematic investigation of bromine monoxide in volcanic plumes from space by using the GOME-2 instrument, Atmospheric Chemistry and Physics, 13, 4749–4781, doi:10.5194/acp-13-4749-2013, 2013.

Kern, C. and Lyons, J.: Spatial distribution of halogen oxides in the plume of Mount Pagan volcano, Mariana Islands, Geophysical Research Letters, 45, 9588–9596, doi.org/10.1029/2018GL079245, 2018

Lübcke, P., Bobrowski, N., Arellano, S., Galle, B., Garzón, G., Vogel, L., and Platt, U.: BrO/SO2 molar ratios from scanning DOAS measurements in the NOVAC network, Solid Earth, 5, 409–424, doi:10.5194/se-5-409-2014, 2014.

Theys, N., De Smedt, I., Yu, H., Danckaert, T., van Gent, J., Hörmann, C., Wagner, T., Hedelt, P., Bauer, H., Romann, F., Pedergnana, M., Loyola, D., and Van Roozendael, M.: Sulfur dioxide retrievals from TROPOMI onboard Sentinel-5 Precursor: algorithm theoretical basis, Atmos. Meas. Tech., 10, 119-153, https://doi.org/10.5194/amt-10-119-2017, 2017.

---

## Author Comment (AC1) · 17 May 2019

**Response to Reviewer 1**

We thank the reviewer for their in depth review of the manuscript which has resulted in a much improved revision. The main comment on the formulation of the forward model has been addressed with a significant re-write of the description of the fitting procedure, we hope this is now sufficiently clear. Other significant changes include a comparison with the QDOAS software instead of a custom written DOAS code and the addition of data from a scanning spectrometer station to emphasise the issue of contamination.

**Major Comments**

*In the Abstract (l. 5) the authors claim to present a novel technique using intensity fitting and a solar reference. This claim, in my opinion, is not fulfilled because, in general, intensity fitting is equivalent to fitting in optical density space and commonly applied in the IR community. Secondly, using a literature solar reference was already applied by Salerno et al., 2009b and discussed more thoroughly by Lübke et al., 2016. What makes the described approach interesting is the calculation in high resolution. This detail, however, is not motivated enough in the abstract/manuscript. Why should it be advantageous to fit in high resolution while the actual measurement spectrum does not provide any information in this increased resolution anyway?*

*I would like to take this opportunity to mention that there are numerous publications addressing the correction of simplifications introduced by DOAS. These were compiled by Referee #1 in his/her comment on Burton and Sawyer, 2016. The discussion from the Referee comment should be included in the introduction or the discussion of the manuscript.*

The abstract, introduction and conclusion sections have been changed to reflect the advantages of fitting in high resolution rather than solely on fitting in intensity. We acknowledge that fitting in intensity is not new in of itself, however we respectively disagree with the statement that fitting in intensity is the same as fitting in optical depth due to how factors like the $I_0$ effect are handled.

We have included a more in-depth discussion of the corrections applied to DOAS retrievals (p. 2 l. 15 – 31).

*The method section states that multiplications and convolutions were noncommutative (p. 4, l. 29). This is wrong, because a convolution is an integral, which is linear (see https://en.wikipedia.org/wiki/Integral). Therefore, all conclusions drawn from this hypothetical non-commutitativeness are also wrong. In DOAS, the convolution is therefore not required to be applied in every fit iteration (p. 5, l. 1). Please revise the manuscript an remove implications of this erroneous assumption.*

Thank you for highlighting this fact, we have removed this from the manuscript.

*The forward model (Eq. (10)) approximates broadband absorption with a polynomial factor P(λ) in intensity space. However, Eq. (3) suggests that it should be rather exp(P(λ)). What is the influence of this approximation on the fit result?*

The description of equation 10 (now equation 9 in the updated manuscript) has been expanded to explain why a polynomial had been used instead of the exponential of a polynomial (section 2.2.1).

The polynomial takes into account various factors including the Mie and Rayleigh scattering and transmission functions of the optics used.

*In my opinion, the formulation of the inversion problem is not completely clear and should be improved (Sect. 2.3.2). - Please detail, which parameters are fitted (or included in the state vector), eventually in an additional table. Please also include the initial values. - How are stretch and shift treated? They do not appear in Eq. (10). Is the fit applied on filtered spectra as stated in p. 4, l. 18f? - In my opinion, a real advantage of fitting in high resolution is the opportunity of retrieving an actual ILS parametrisation simultaneously. Has this option be considered? If not, why? – Figure 2 is a bit confusing. It seems that the "Final Spectrum" is entirely calculated before fitting. Please specify, which steps are done before fitting and which parameters are included in the fit. The fit is usually an iterative process, which could be sketched in the figure. - What are the termination criteria of the fit procedure?*

The forward model has been expanded upon (now section 2.2.3) and more detail in how the fit is achieved has been added, hopefully this is now clear. Also figure 2 has been updated to improve clarity.

Retrieval of the ILS simultaneously with the fit is possible and has been tested, although not rigorously and so was not included in the manuscript.

*The iFit results are compared to classical DOAS results in Sect. 3.3. The description of the applied DOAS method, for which the reader is referenced to Sect. 2.2 (p. 10, l. 5), is not complete. For instance, it is not clear, which software was used. From the provided description one may conclude that did not apply a commonly used and validated implementation like QDOAS. In my opinion, a standard software package should be used in order to provide a meaningful comparison between iFit and DOAS.*

For this comparison a DOAS script was written in Python and used to fit the spectra. This has now been changed and the comparison is now with the QDOAS software.

*Bobrowski et al. 2010 state a "standard evaluation range of approximately 310 to 325nm". What is the effect of shifting the range for the DOAS retrieval to 305-318 as applied in this study? I would assume that interferences from $O_3$ absorption and Rayleigh scattering are more dominating at shorter wavelengths. Please discuss this issue in the revised manuscript.*

The lower wavelength range was chosen as it seemed to perform best for iFit, however after the comments made by both referees all analyses were repeated using the 310 – 320 nm window to be more in line with the common analysis windows. While the reported uncertainty increased (due to the reduced strength of the $SO_2$ spectrum) it was found that this range was just as robust as the lower window for iFit. There was also little change in the results from the previous DOAS analysis by shifting the fit window, although this has now been replaced with the QDOAS analysis.

*The results of the comparison between iFit and DOAS are compiled in Fig. 11. The discussion of the comparison, however, is a bit meagre. For example, Fig. 11 reveals a bias compared to the background (and the DOAS) value at the left and right tail. The reference was taken at 9:59, which is approximately in the centre of Fig. 11 (a). At 10:05, however, iFit results seem to be significantly positively biased. Please discuss this issue in the manuscript. Furthermore, a quantitative comparison between iFit and DOAS like compiled in Table 1 would be favourable.*

The reanalysis using QDOAS has removed this bias, and additional descriptions of the comparison have been added, including a discussion of the reported errors by both iFit and QDOAS (p. 10 l. 6 - 9).

*When treating the $I_0$ effect and strong absorption, the knowledge about the ILS is crucial. It seems that a Gaussian ILS was applied throughout the study (p. 7, l. 27). The paper by Beilre et al., 2017 investigates parametrisations of different spectrometers and states, that the Gaussian may be a sufficient ILS parametrisation only for some instruments. Therefore, I suggest to add a plot showing the measured ILS for the applied instruments.*

In the reanalysis a super-Gaussian is used in the stead of the Gaussian used before and this has been found to represent the measured ILS more accurately, for which we thank the reviewer. Figure 4 shows the ILS for each spectrometer has been added, and a fitted super-Gaussian function is used throughout the analysis, the parameters of which can be found in table 2. Thank you for bringing this to our attention.

*I am a bit confused about the discussion of the computational speed (Sect. 3.4). Actually, iFit requires a convolution in every fit step and, therefore, an inferior performance of iFit compared to DOAS can be anticipated a priori.*

*As a comparison, Beirle et al. (2013) claim to be able to process non-linear DOAS analysis using 0.004 seconds per fit with DOASIS. When they implemented their linearised method in MATLAB, they even achieved 0.00004 seconds per fit. In the study here, a faster CPU with 3.2 GHz (instead of 2.5 GHz) was applied and achieved not less than 0.09 seconds per spectrum. Hence, iFit is more than a magnitude slower than DOASIS on a slower PC.*

*Furthermore, the stated observation that changing the initial value could speed up the process by more than a factor of 2 indicates that either the algorithms converges very slowly or that the standard initial values are not chosen optimally. Please discuss. In my opinion, computational speed should not be overrated for the application of a scientific algorithm. If the scientific question requires a slower data evaluation algorithm, it shall be favoured over a faster and less accurate one. The algorithm presented here works without a measured reference spectrum, which is a quality of its own. Therefore, I suggest to remove the discussion about computational speed from the manuscript altogether (also: p. 10, l. 28 "We have...")*

This section was included after a comment from a previous reviewer of the Burton and Saywer (2016) manuscript, however we agree with the reviewer's comment and so have now moved it to the appendix. The comparison of the computation speed of iFit and DOAS will depend on how the analysis programs are written and which language they are written in. Python was chosen as it is Open Source and contains many useful libraries that can be utilised, but it can be slower than other languages. For the comparison between DOAS and iFit a similar code was written to analyse the spectra using the DOAS methodology to produce a test between the methods, not the choice of programing language.

*I do not agree with the author's statement that fitting in intensity space is more intuitive (p. 11, l. 1). This is a personal opinion and should be omitted. For me, absorption is an asymptotic process, which is intuitively linearised by transforming from intensity space to optical density space.*

This comment was based on our experiences working with observatory staff, for whom the iFit program was originally designed. We agree that this is not a fact but an opinion, and so it has been removed.

**Minor comments**

*I am a bit confused about the identity of the first author. Is his first name Ben or Benjamin? Personally, I would refrain from using nicknames in scientific author lists and affiliations.*

The birth name of the first author is Benjamin, but the name he has used in all professional correspondence, where given the choice, is Ben.

*p. 1 l. 8 "number of advantages" -> "primarily" is a bit vague. Please be more specific.*

These statements have been removed.

*p. 1, l. 10 Remove the repeating "without the use of a reference spectrum"*

Removed

*Please avoid qualitative statements like p. 1, l. 9: "accurately" p. 1, l. 10: "well suited" p. 1, l. 11: "strong potential" p. 7, l. 20: "accurately" p. 10, l. 9: "very well" p. 10, l. 24: "accurately" p. 11, l. 6: "accurately" and use quantitative statements instead.*

These statements have been removed.

*Sect 2.2.2 Please add a statement that the $I_0$ effect is due to the non-commuting of exponential function and convolution. This effect can be corrected for in DOAS evaluations.*

This has been included (p. 4 l. 4)

*Eq. (8): The clarity of this formula could be improved, e.g. by omitting "(λ)" and some more brackets. Maybe also use \frac{}{} as in Eq. (A3) in the cited Aliwell et al. 2002.*

Corrections made to formula (now equation 6)

*p. 5, l. 26 omit "another"*

This has been removed.

*p. 10, l. 31: Please remove the sentence starting with "Remaining ..." because it is not based on findings in the paper. All effects mentioned can be addressed by DOAS.*

This statement has been removed.

*p. 11, l. 3: This is not a particular improvement of iFit, because low intensity issues are visible in raw spectra. Please omit this statement.*

This statement has been removed.

*Figure 4(d): Caption and legend are not matching. What is the difference between measured and synthetic Ring? Please clarify.*

The legend has been corrected. The difference was explained in the text (p. 8 l. 5 of previous manuscript, p. 8 l. 2 in the revised manuscript).

*Caption of Fig. 11: Please detail the definition of CST time.*

CST is a time zone, this has been clarified in the figure caption (now figure 13).

*Please add gridlines to all plots to improve readability.*

These have been added.

*Please add subplot labels like (a), (b) and so forth to Figs. 1, 6, 7, 8, and 9 for better reference.*

These have been added.

*Please arrange the subplots in Figs. 4, 5, and 10 first left to right then top to bottom.*

This has been done.

---

## Author Comment (AC2) · 17 May 2019

**Response to reviewer 2**

**No comprehensive description of their method is provided**

The section describing the fitting procedure (section 2.2.3) has been re-written and expanded upon, we hope that it is now clear.

*Page 6, lines 27-29: "A model spectrum is then built on the high resolution model grid, which typically has a spacing of 0.01 nm..." How is this model spectrum build? By a convolution with a Gaussian ILS with a FWHM of 0.01 nm?*

The model grid is simply a 0.01 nm spaced grid onto which all reference spectra are interpolated by cubic spline. The model spectrum is the result of inputting the fitted parameters into equation 9. The fitting process is now described in more detail, and hopefully more clarity, in section 2.2.3.

*How are the effects of the dark current and "bias" determined and corrected? Remark: the latter is typically called "offset" rather than "bias" (see e.g. Platt and Stutz, 2008).*

This is explained in section 2.2.3. Thank you for highlighting that fact, we now refer to it only as the offset signal.

*Flat spectrum: Esse et al. retrieved the "flat spectrum" by a simple averaging. In contrast, Lübcke et al. (2016) retrieved the instrument effects (flat spectrum but also further instrument effects such as temperature effects) by a Principal Component Analysis (PCA). For me, the PCA approach appears to be more comprehensive. See also my comment 2.3. Please motivate the choice of a simple averaging instead.*

The choice of averaging was to directly measure the instrument properties rather than retrieve them from measurement data. This was chosen as it does not rely on selecting measurement spectra for a training set, rather it can be premeasured in the lab before measurements take place. Other instrument effects could remain and it would be possible to extract these separately from the flat spectrum using a similar method to Lübcke *et al* (2016) but this would remove one of the main benefits of the iFit method. This has been explained in the text (p. 6 l. 1).

*Background polynomial: Is equation (10) correct? Physically, all light attenuation effects are on the same footing, i.e. both absorption and scattering effects are summands in the argument of the exponential function. In principle, the scattering effects can of course be written in the presented way e.g. as P(λ) = exp(P\*(λ)) where P\*(λ) is the "real" broad-band scattering polynomial. But this is strictly different from the polynomial as it is used in DOAS. In particular, its coefficients will be different to the polynomial P(λ) denoted in equation (3). Please clarify.*

The discussion of the background polynomial has been expanded to clarify this point (p. 5 l. 11). A single polynomial function is used to account for a number of effects including the Mie and Rayleigh scattering as well as the transmission of the optics.

*Ring spectrum: First, I have analogous doubts concerning R(λ) in equation (10) and equation (9). Second, please make more explicit how is the Ring spectrum retrieved.*

The Ring spectrum was correctly applied in the forward model, but we agree that the description was inconsistent between the two equations. This has now been updated. The ring is retrieved as part of the overall fit (p. 6 l. 18 and table 1).

***Sky spectrum: How is it constructed? By adding the absorption effects of background $O_3$ and $NO_2$? If yes, how is the correct background amount determined?***

This description of the fitting process was perhaps confusing, and so has been changed. Hopefully it is now clearer. All parameters given in table 1 are fitted simultaneously.

***Plume spectrum: Is also $O_3$ included in the fit step from sky spectrum to plume spectrum? If not, how are the diurnal variations of stratospheric $O_3$ contributions and assessed and corrected? Furthermore, why is particularly BrO (but no other gases) included in the $SO_2$ fit scenario? The BrO absorption is rather negligible in the typical SO2 fit ranges.***

As above this comment stems from the description of the algorithm, which was perhaps confusing. $O_3$ is included in the fit to account for the changes in path length throughout the day. Fitting the BrO spectrum is indeed redundant at these wavelengths, and so has been removed when reanalysing the data.

***Instrument line shape function ILS: "The ILS was measured using a mercury lamp to be 0:50 (±0:01) nm" (page 8, line 27). The uploaded data does provide ILS (only) at 302nm (instrument H15972) with a FWHM of 0.58nm and at 301nm (instrument FLMS02101) with a FWHM of 0.60 nm. Please provide the full mercury spectra for a presented instruments. (Remark: both uploaded ILS have indeed an about Gaussian shape. A super-Gaussian model proposes exponents of (only) 2.3 and 2.1 and the asymmetry is rather small as well.)***

***The ILS is a unique property of the instrument, although it varies in general with temperature and wavelength. Accordingly, for a given instrument (and similar temperature) all convolution operations have to use one identical ILS. Applying different ILS can cause a significant decrease in accuracy. Furthermore, all compared spectroscopic should apply an ILS retrieved at the same wavelength in order to be consistent. Ideally, this wavelength is chosen in the wavelength range, e.g. at 315 nm. For practical reasons, the mercury line either at 302nm or at 334nm should be chosen. Please clarify which measurement results for the ILS are used. I propose to add a table to the manuscript which lists all instruments used in this study and their spectroscopic properties.***

***However, later a modelled ILS with "an ILS width of 0.56 nm" (page 10, line 9) has been used instead of the measured ILS. Please clarify why instead of the exact measurement results an apparently wrong ILS is used at this step.***

One change to the updated manuscript was to replace the Gaussian ILS with a super-Gaussian. Figure 4 now shows the Hg spectra and fitted 302 nm line and the instrument properties are summarised in table 2. The fitted super-Gaussian ILS is now used for all analyses.

***How are the wavelength shift and stretch determined and corrected?***

The wavelength shift and stretch are fitted as part of the forward model. This has been clarified in the text (p. 6 l. 18).

*Furthermore, physical-logical the wavelength shift and stretch should be actually applied on the measured spectrum in order to correct for temperature-driven variations of the instrument during the measurement. Analogously, the "flat spectrum" should be applied on the simulated spectrum in order to correct for the instrument effects of the real instrument. I can imagine that these inconsistencies are mathematically identical and may thus lead to the same results. Please clarify why/whether these inconsistencies are required.*

We disagree with this statement, as to include the shift in the forward model as a fitted parameter it needs to be applied to the model grid, not the measurement grid. The shift could be predetermined and applied to the measurement, however it was found that fitting it was more robust. The flat correction should be applied to the measurement as it is a property of the instrument pixels, not the wavelength. This has been emphasised in the text (p. 6 l. 15).

*What actually does "perform fit"? Is it an ordinary DOAS fit? Or is it iteratively minimising $\tau = log\left(\frac{I_{right\ hand\ side}}{I_{left\ hand\ side}}\right)$ by means of varying the SO2 column density?*

The measured intensity spectrum is fitted by the forward model by minimising the residual between the two (in intensity). The fitting algorithm has been described in more detail in the text and is now hopefully clearer (p. 6 l. 23).

*Is a stray light correction applied?*

Yes, using the intensity between 280 and 290 nm. This is explained in the text (p. 6 l. 13).

**Missing comparison with the state of the art**

*The manuscript is motivated by the possible underestimation of the $SO_2$ slant column density in a volcanic gas plume. However, no iFit results for such a scenario have been provided. Please explain this inconsistency.*

Data from the ENIC scanning station of the FLAME network has now been included as an example when this is the case (section 4.2.2). From this a clear underestimation due to the contamination can be seen.

*According to the list of literature provided in the manuscript (and to my knowledge), the approach from Lübcke et al. (2016) is the current state of the art to face such background contamination issue. I highly recommend a direct quantitative comparison of these two methods when applied on the same data (ideally contaminated data) in order to reveal the major differences.*

Although we see the merit of a direct comparison, we believe that this would be beyond the scope of this manuscript as it would require developing a full analysis routine for the method provided by Lübcke *et al.* (2016). We believe that the advantage of having a "point and shoot" retrieval method is a significant advantage when there is no large training dataset with which to perform the PCA. Conversely, the method presented by Lübcke *et al.* (2016) could be preferable when access to the spectrometers is not possible, and so the flat spectrum cannot be characterised. These points have been further emphasised in the text (p. 10 l. 23).

*Esse et al. propose to use iFit for evaluating data recorded by permanent monitoring stations. Monitoring stations are typically not temperature controlled in order to improve their robustness*

*and to lower their power consumption (see e.g. Galle et al., 2010). iFit has been tested exclusively for temperature stabilised instruments and Esse et al. concluded that "care must be taken if the spectrometers are not temperature stabilised" (page 11, line 9). Accordingly, Esse et al. have not provided evidence that iFit is suitable for monitoring stations. This is in particular in sharp contrast to the approach from Lübcke et al. (2016) which presented their results for contaminated data from monitoring stations.*

None of the measurements shown here were temperature controlled. We also now present additional spectra from the FLAME network on Etna showing contaminated data (section 4.2.2).

*I agree with Esse et al. that also a comparison with standard DOAS (recorded background spectrum) appears to be mandatory. For the arguments stated in the very first paragraph, I expect that a standard DOAS approach performs in general (i.e. for non-contaminated scenarios) better than iFit. In contrast, the narrative in the current manuscript is rather one-sided, highlighting possible problems in the DOAS approach only. I highly recommend that any subjective valuations are neglected from the technical manuscript parts (e.g. "methods", and "results"). Differences between iFit and DOAS should be discussed later in the "discussion" part where all evaluating statements should be supported by (quantitative) evidence.*

I agree with this statement that where an uncontaminated reference spectrum is available and the appropriate corrections (e.g. $I_0$ effect) are applied then the standard DOAS algorithm will outperform iFit as iFit implements a relatively simplistic radiative transfer model to fit the measured spectra. This is shown and commented in the comparison between iFit and DOAS (p 10 l. 6).

*Esse et al. conclude "the lack of a requirement for a reference spectrum means that iFit would be especially well suited to deployment in permanent scanning stations" (page 11, line 14). Permanent scanning stations scan typically from horizon to horizon and thus automatically recorded the reference spectrum. Applying iFit thus does not provide any gain in measurement time in particular at permanent scanning stations. Furthermore, probably only few permanent measurement stations are at all affected by background SO₂ contamination. Accordingly, possible benefits from iFit are limited to those stations. Please limit your conclusions with respect to those scenarios where you can provide evidence that iFit at least does not perform more poorly than the alternative approaches.*

We have adjusted this statement to reflect the fact that iFit favours automated analysis as no definition of a reference is required. This is not a comment on the time taken to perform the scan, but the removal of potential sources of error. We agree that not all stations will often suffer from contamination, but would like to emphasise that all stations could (and likely will at some point). Therefore methods which use a synthetic reference are preferable.

*Page 4, lines 11-19: Is there any need to discuss the option of a high-pass filter? Is a high pass filter used in iFit or for the DOAS retrieval in this manuscript? If not, this paragraph appears to be redundant. The figures 4, 5, 10 show results of the total absorption cross section.*

The high pass filter was only included to simplify the equations – however we agree that it could confuse the reader as both iFit and DOAS analyses presented do not actually use one. It has now been removed and a broadband polynomial included in the equations.

*Page 4, line 26: "The ILS can either be a mathematical function (such as a Gaussian)..." The ILS is a property of the instrument and has in general an arbitrary shape. Although it can be indeed often approximate in good agreement by a Gaussian line shape function, the real ILS itself is not a mathematical function!*

This was a poor choice of wording, and has been changed in the manuscript to state that it is a property of the instrument but can be approximated with a mathematical function.

*Page 4, lines 28-31: Equation 6 is not true! The convolution operation and the scalar multiplication operation are commutative!*

Thank you for highlighting this error, it has been removed from the manuscript.

*Page 6, lines 29-30: "A wavelength-shift is a common correction in DOAS...". This is correct, however, when the spectra are wavelength-calibrated prior to the DOAS fit this shift is typically in the order of 0:001 nm. Do Esse et al. refer to the additional wavelength shift parameter which is usually allowed between the measurement spectrum and the absorption cross-sections in order to partially compensate for the convolution with a (slightly) wrong ILS? Anyway, this wavelength shift is typically limited to ±0.2nm rather than ±2 nm. A wavelength shift of 2nm appears to be absurdly large. Please clarify.*

The 2 nm padding is included for two reasons: firstly to accommodate the wavelength shift (which as the reviewer points out is typically small) and secondly to avoid the edge effects incurred by the convolution of the model spectrum with the ILS. This has been clarified in the text (p. 6 l. 19). Typical retrieved shifts are of the order of 0.1 nm.

*Page 7, line 27: "The wavelength region used for these results (304 - 320 nm) is common to most scattered sunlight retrievals of SO2." This is not true. In DOAS - the predominant remote sensing technique for volcanic SO2 - the used wavelength range starts almost exclusively at 310nm (e.g. Lübcke et al., 2016), 312nm (e.g. Theys et al., 2017; Kern and Lyson, 2018), 314nm (e.g. Lübcke et al., 2014; Dinger et al., 2018), or 326nm (e.g. Hörmann et al., 2013). The reason for these lower limits is, that the applied approximation in DOAS are only justified as long as the "absorbance" is not much above 0.1. For the data presented in Figure 10, this means the DOAS retrieval should start not lower than at 314 nm. This limitation does not have to hold for an intensity based fit, however, Esse et al. have to make sure that all presented DOAS data are retrieved for an absorbance below 0.1. Otherwise their DOAS results would be too low and a quantitative comparison between iFit and DOAS therefore flawed.*

This comment was meant to reflect the general wavelength region commonly used (e.g. near to the absorbance features of $SO_2$) not the specific wavelengths used in retrievals, but we can see how it is misleading and so this phrase has been clarified. As already mentioned we also now perform the fits between $310 - 320$ nm which is more in line with standard DOAS retrievals.

We also realise that the displayed absorbance features on figures 4, 5 and 10 were actually optical depth (as described in the text) not absorbance, we apologise for this error. The figures have been updated. The peak absorbance value during the traverses was 0.12, but typically it was ~ 0.07 in the plume.

*Page 8, line 31: "In particular, use of higher wavelengths leads to an overestimation in the retrieved SO2, possibly due to the reduced strength of the $SO_2$ absorption spectrum at higher wavelengths". This interpretation of the findings is not supported by further evidence. Furthermore, the "reduced strength of $SO_2$ absorption" can be expected to result in a larger fit error but there is no obvious reason why this should cause a less accurate result. In fact, I would interpret the findings other way round: the lower the wavelength the larger is the underestimation in $SO_2$ due to saturation effects and a decreasing solar background radiation (see Platt and Stutz, 2008). At least for DOAS the "absorbance" should be below 0.1 in order to keep the applied approximations justified. With these fundamental limitations in mind, I consider the results for the wavelength range from 310-320nm the most accurate. Furthermore, I expect that starting at 314nm or 326nm would give larger and even more accurate results in particular for the cells above 500 ppmm. In consequence, iFit would overestimate the $SO_2$ slant column density. Please provide evidence for your interpretation. In particular, I highly recommend to present (additionally or exclusively) DOAS results when the wavelength range starts at least at 310 nm.*

The discussion of the cell data have been rewritten. For iFit use of higher wavebands results in a large offset in the retrieved $SO_2$ SCD due to the program supplying an $SO_2$ SCD to fit features in the residual. This discussion has been included in the updated manuscript (p. 8 l. 22).

**Some minor formal objections**

*Inconsistent use of brackets (e.g. compare equation 7b and equation 8)*

This has been corrected to ensure the equations are consistent.

*The (slant) column densities are sometimes denoted by $a_i$ and sometimes by $\alpha$. They have to be denoted by the same consistent letter throughout the manuscript. Furthermore, they should always hold the index.*

The column densities are now all referred to as $a_i$ with the index.

*$I_0$ and $I^*_0$ appears in several forms throughout the manuscript. They should be consistently denoted by a strictly constant sign*.

This has been corrected.